

# Celecoxib treatment alleviates cerebral injury in a rat model of post-traumatic epilepsy

Lei Chen[1,*], Qingsheng Niu[2,*], Caibin Gao[2] and Fang Du[3]

[1] Department of Neurosurgery, The First People's Hospital of Shizuishan, Shizuishan, Ningxia Hui Autonomous Region, China
[2] Ningxia Medical University, Yinchuan, Ningxia Hui Autonomous Region, China
[3] Emergency and Critical Care Center, Hubei University of Medicine, Shiyan, Hubei, China
[*] These authors contributed equally to this work.

## ABSTRACT

**Background**. An important factor contributing to the development and occurrence of post-traumatic epilepsy (PTE) is neuroinflammation and oxidative stress. The effects of celecoxib include inhibiting inflammatory reactions and antioxidant stress and reducing seizures, making it a potential epilepsy treatment solution.

**Objective**. To observe the effect of celecoxib on early epilepsy in post-traumatic epilepsy rats. Methods: Twenty-four adult healthy male Sprague-Dawley rats were randomly assigned to three groups: sham-operated, PTE, and celecoxib. A rat model of PTE was established by injecting ferrous chloride into the right frontal cortex. Afterward, the behavior of rats was observed and recorded. 3.0T superconducting magnetic resonance imaging (MRI) was used to describe the changes in ADC values of the brain. HE and Nissl staining were also used to detect the damage to frontal lobe neurons. Furthermore, the expression of COX-2 protein in the right frontal lobe was detected by Western blot. Moreover, the contents of IL-1 and TNF-$\alpha$ in the right frontal lobe were detected by enzyme-linked immunosorbent assay.

**Results**. Compared with the PTE group, the degree of seizures in rats treated with celecoxib declined dramatically ($P < 0.05$). Celecoxib-treated rats had significant decreases in tissue structural damage and cell death in the brain. The results of the MRI showed that celecoxib reduced the peripheral edema zone and ADC value of the cortex around the damaged area of the right frontal lobe in the celecoxib-treatment group, which was significantly decreased compared with the PTE group ($P < 0.05$). Furthermore, celecoxib decreased the expression of COX-2, IL-1$\beta$, and TNF-$\alpha$ in brain tissue ($P < 0.05$).

**Conclusions**. In PTE rats, celecoxib significantly reduced brain damage and effectively reduced seizures. As a result of celecoxib's ability to inhibit inflammation, it can reduce the edema caused by injury in rat brain tissue.

Corresponding author
Fang Du, dfshenhaiyu@163.com

## INTRODUCTION

An epileptic seizure after traumatic brain injury (TBI), also known as post-traumatic epilepsy (PTE), is caused by a brain injury, such as trauma, a brain tumor, a stroke, or an infection (*Piccenna, Shears & O'Brien, 2017*). Various mechanisms have been suggested as causing post-traumatic epilepsy, including increased inflammatory markers, altered blood–brain barrier, astrocyte changes, and glucose metabolism dysregulation (*Webster et al., 2017*; *Sharma et al., 2019a*; *Sharma et al., 2019b*). Despite intensive research, definitive tests and effective treatments are lacking, as epileptogenic factors associated with PTE are incompletely understood mechanistically. Evidence from *in vivo* experimental models suggested a rapid release of inflammatory cytokines and oxidative stress in peritraumatic brain tissue, a consequence of epilepsy contributing to its pathophysiology (*Rawat et al., 2019*). Thus, anti-inflammatory therapy may be of therapeutic value for preventing PTE.

Much attention has been focused on the cyclooxygenase-2 (COX-2) enzyme because of its central role in the development of epilepsy and seizure generation ((*Rojas et al., 2014*)). Previous studies have also reported that COX-2 further increased the production of proinflammatory mediators to aggravate seizure severity (*Shimada et al., 2014*). From clinical practice and research, NSAIDs that inhibit the COX-2 enzyme have been developed due to their strong anti-inflammatory and analgesic effects (*Zarghi & Arfaei, 2011*). Celecoxib, a COX-2 selective inhibitor, has been designed to reduce pain and inflammation. Because of its phenyl sulfonamide moiety, celecoxib has a higher selectivity of COX-2 than rofecoxib and aspirin, revealing a stronger anti-inflammatory effect (*Sharma et al., 2019a*; *Sharma et al., 2019b*). There is also extensive evidence from experiments in animal models that celecoxib can reduce brain prostaglandin E2 expression, oxidative stress, apoptosis of nerve cells (*Wang et al., 2018a*; *Wang et al., 2018b*), and induce nerve repair after nerve injury. However, there is currently no study to investigate the therapeutic effect of celecoxib on PTE. Hence, in this study, we explored the therapeutic effect of celecoxib on brain injury in PTE rats, which is associated with the anti-inflammatory effect of celecoxib.

## MATERIALS & METHODS

### Reagents

Iron (II) chloride (Ferrous chloride or $FeCl_2$) was obtained from Sigma Aldrich (St. Louis, MO, USA), 0.2mmol/L $FeCl_2$ solution was freshly made before use by dissolving $FeCl_2$ in normal saline. The celecoxib solution was prepared by adding celecoxib to normal saline and freshly made before use.

### Animal preparation and the establishment of the PTE model

Twenty-four adult healthy male Sprague-Dawley rats (7–8 weeks old, 240 ± 30 g) were obtained from the Experimental Animal Center of Ningxia Medical University (License No. SCXK (Ning) 2020-0001). All rats were kept in the Animal Center of the Brain Laboratory of Ningxia Medical University at a uniform temperature of 20−22 °C and on a light/dark cycle of 12 h with free access to food and water. Study animals were cared for and used in accordance with the Laboratory Animal Care and Use Guidelines. Shizuishan First People's

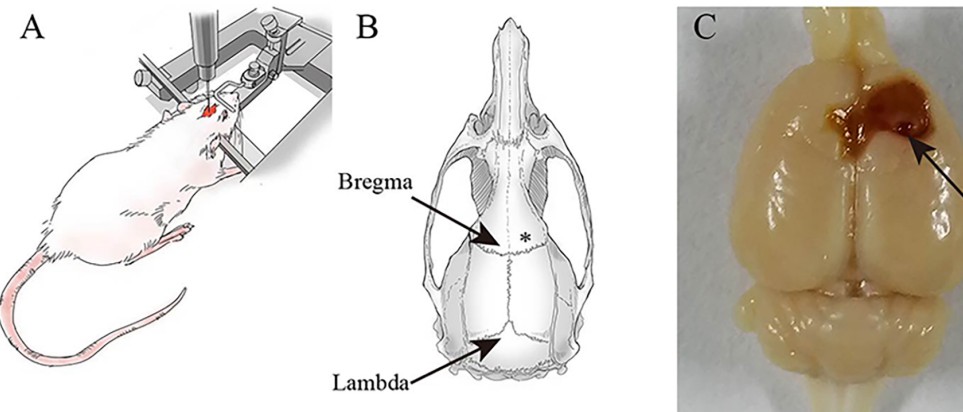

**Figure 1** **Establishment of post-.traumatic epilepsy model.** (A) A rat is fixed on a stereotaxic device. (B) FeCl2 injection site marked with an asterisk (*). (C) The brains are excised on the 7th day after FeCl2 injection. Arrow: Pinhole.

Hospital's Institutional Animal Care and Use Committee approved the animals used in this study. The animals, anesthetized by intraperitoneal injection of 1% Pentobarbital Sodium (40 mg/kg), were mechanically fixed on the brain stereotactic apparatus and with the skull exposed. Then, we lowered the needle to right above the skull to visualize where the hole needed to be drilled. As shown in Fig. 1, after stereotactic coordinates of the rat brain were found, we targeted FeCl2 into the right frontal lobe using the coordinates: $M/L = +/-3.00$ mm, $A/P = 2.00$ mm with an injection rate of 1 $\mu$L/min and the injection volume of 5 $\mu$L. Racine scores served as a quantitative way to categorize the intensity of a seizure that an epileptic patient experienced, including five symptoms: mouth and facial movement, head nodding, forelimb clonus, rearing with forelimb clonus and hind legs (*Racine, 1972*). A successful epilepsy model was defined as a rat's epilepsy grade reaching the Racinestandard grade III or above. Furthermore, epileptic seizures were monitored continuously for 7 days after the successful construction of the epilepsy model.

## Experimental procedure

The animals were randomly assigned to the following three groups ($n = 8$ in each group): (1) Sham-operated group, the rats were injected with an equal volume of normal saline alone. (2) PTE group, the rats were injected with 5 $\mu$L FeCl2 and underwent epilepsy (*Zhang et al., 2022*). (3) Celecoxib group, the animals were administered celecoxib (20 mg/kg, dissolved in 0.9% normal saline) by gavage at the same time every day for 7 days after the successful construction of the epilepsy model. At the end of the 7 days, rats were euthanized using intraperitoneal injection of 800 mg/kg pentobarbital sodium, and the brain tissues of the rats were harvested. In addition, rats were euthanized by intraperitoneal injection of 800 mg/kg pentobarbital sodium at the end of the experiments or when reaching human endpoints if they demonstrated signs of weight loss, appetite loss, depression, anxiety, infection, organ failure or if they became moribund.

## Magnetic resonance imaging (MRI) scan

On the 3rd and 7th days after the model was delivered, MRI scanning (GE Signa architect 3.0T superconducting MRI scanner) and an 8-channel brain coil (Shanghai Chenguang Medical Technology Co., Ltd.) were performed to scan the rat brain. The head of the rats was fixed in the center of the coil with autonomous breathing and light narcosis. The scanning parameters of TSE-T1WI: TR = 470ms, TE = 10 ms, layer thickness = 2 mm, interval = 0.2 mm, FOV = 8.0 mm × 8.0 mm, matrix = 128 × 128, NSA = 2; those of TSE-T2WI were as follows: TR = 2500 ms, TE = 85 ms, layer thickness = 2 mm, interval = 0.2 mm, FOV = 250 mm × 250 mm, matrix = 128 × 128, NSA = 2. Then, DWI scanning was conducted, and the scanning parameters were as follows: TR = 3420 ms, TE = minimum, layer thickness = 2 mm, interval = 0.2 mm, isotropy, scanning number of layers = 12, FOV = 8.0 mm ×8.0 mm, matrix = 128 × 128. T2WI image was used for registration with an atlas by the built-in T2WI software in the MRI scanner to evaluate the two regions of interest (ROI) (1.0−2.0 mm2 (including 91 pixels) where brain injury occurred. When the T2WI interpretations of two senior MRI technicians did not agree, a third MRI diagnostician was required to resolve the disagreement. From DWI, an ADC image was generated at the site of the lesion, and ADC values were then measured in the layer of the primary lesion, the layer with uniform parenchymal composition, the layer with cysts, and the layer with necrotic cells. Furthermore, the diagnostic accuracy of brain injuries was explored in terms of injury area, ADC value, and DTI characteristics (including the FA value, RA value, and VR value). The observations were made on the brain injury boundary and ADC value on the adjacent brain injury area.

## Histopathologic evaluation

Brain tissues of the rats were exercised on the 7th day after PTE. The brain tissue sample was fixed in 4% paraformaldehyde, partially embedded in paraffin, and stained with hematoxylin-eosin and Nissl stained. At least 3 random sections were selected to stain with hematoxylin–eosin and Nissl to observe the histological changes and were examined under an optical microscope in a blinded manner (*Hegazy & Hegazy, 2015*).

## Western blot analysis

COX-2 protein expression was assessed by Western blot analysis. To the harvested tissue, 0.3 mL RIPA lysate containing 1% PMSF and 1% phosphatase inhibitor was added. The homogenized tube was put on ice for 30 min, followed by 15 min of centrifugation at 12,000 g at 4 ° C. The instructions for the BCA kit (Beyotime, Shanghai, China) were followed to determine the protein concentration in the supernatant. Each group was separated using sodium lauryl sulfate-polyacrylamide (SDS-PAGE) gel electrophoresis at 10%, and a PVDF membrane (0.45 mm; Merck Millipore, Darmstadt, Germany) was used to transfer the separated proteins. For 2 h, the membrane was blocked with 5% (wt/vol) skimmed milk powder, and then an anti-cox-2 polyclonal antibody (1:1500, Abcam, Cambridge, UK) was added to the membrane overnight at 4 ° C. Afterward, horseradish peroxidase-conjugated goat anti-rabbit IgG secondary antibodies (Promega, Madison, WI, USA) were added. The protein reaction band was exposed to the film using an enhanced chemiluminescence kit

(Beyotime, Shanghai, China). All experiments were performed in triplicate, and the density of the bands was determined using ImageJ software (*Kartheek & Parameshwaran, 2023*).

## Cytokine measurements

The rat's right frontal lobe tissue homogenate supernatant was removed from the refrigerator. The total protein concentration was detected using the BCA method. The levels of IL-1$\beta$ and TNF-$\alpha$ in brain homogenate were detected by ELISA kit (Elabscience Biotechnology, Wuhan, China) according to the manufacturer's instructions.

## Statistics

Data were presented as each group's mean $\pm$ standard deviation (SD). Racine's scale of rats undergoing PTE was analyzed with repeated measures analysis of variance (ANOVA). In addition, ADC value, ELISA, and Western blot assays were analyzed using SNK-q tests because they showed normal distribution and homogeneity of variance. A $p$-value $<0.05$ was considered significant. Statistical calculations were performed using SPSS22.0 statistical analysis software.

# RESULTS

## Observations on epilepsy

A successful episode started as a simpler seizure, including binocular gaze, whole-body muscle tension, and fine tremor. Then, it became another symptom of seizure, including twitching or stiffening of individual body parts, such as an arm or leg. Consequently, these seizures affected a larger brain region, making rats lose balance and fall. By contrast, rats treated with celecoxib only showed milder forms of seizure such as facial and mild convulsions. In addition, sham-operated rats did not show symptoms of seizures and were slow in action. As shown in Fig. 2, the Racine's scale of rats undergoing PTE showed a decreased trend over time. Furthermore, the celecoxib-treated group showed a significantly reduced Racine's scale compared with the PTE group at the third, fifth, sixth, and seventh day ($F = 29.167$; $P < 0.01$).

## Detailed brain images produced by Magnetic Resonance Imaging (MRI)

MRI was used in the observation of post-traumatic epilepsy rat models. Figure 3 shows high signal intensity on T1WI and low signal intensity on T2WI around the injection site of the right frontal lobe on the 3rd day after the model was delivered. On the contrary, the higher T1 signal intensity and low T2WI signal intensity resulted in damage to the brain's frontal lobe on the 7th day. All of these dates showed lesions of the brain's right frontal lobe and a prominent peripheral edema zone. A significantly reduced damaged site and peripheral edema zone were also observed in rats treated with celecoxib. Moreover, the result showed that the mean ADC value at the lesion was significantly decreased in the PTE group compared to the celecoxib group and sham-operated group ($0.834 \pm 0.143$ *vs.* $1.012 \pm 0.113$ and $1.142 \pm 0.109$; $P < 0.05$) on the 3rd day. On the 7th day, the PTE group showed increased ADC value compared to the sham-operated group and the celecoxib group ($1.291 \pm 0.197$ *vs.* $0.736 \pm 0.122$ and $0.789 \pm 0.060$; $P < 0.05$).

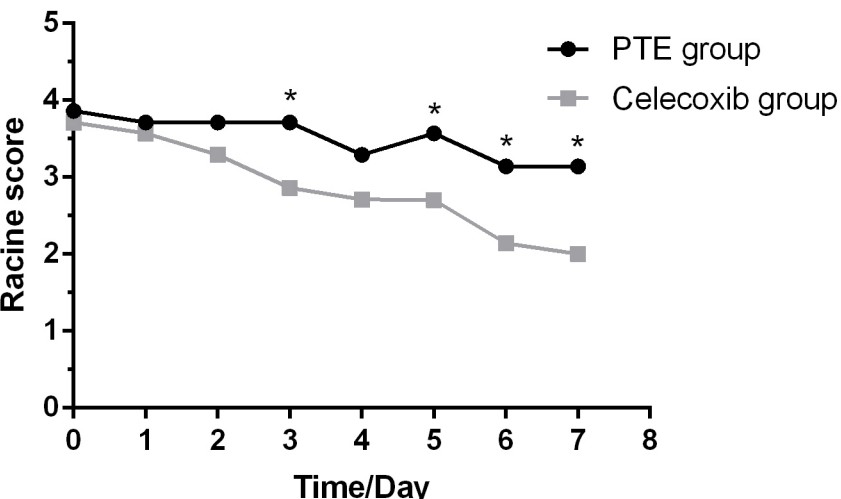

**Figure 2** **Racine score in rats for modeling of post-traumatic epilepsy over 7 days.** Celecoxib group shows reduced Racine scores at different time points compared with the PTE group on the third, fifth, sixth, and seventh day. The Racine scores are expressed as the mean ±SD; $n = 8$ for each group. * $p < 0.05$.

## Histological changes in the brain tissue of each group

Figure 4 shows the histopathological morphology of rats in each group on the 7th day. The rats in the PTE group exhibited reduced neurons, increased intercellular spaces, condensed cytoplasm, shrunken or dissolved nuclei, and swollen glial cells. By contrast, the celecoxib group demonstrated mild cellular edema, relatively increased neurons, and an abundance of highly purified microglia. As expected, the sham-operated group showed normal rat brain tissue, such as intact neurons and glial cells. In addition, Nissl staining images showed that the number of nerve cells in the PTE group was significantly less than in the celecoxib and sham groups.

## Expression of COX-2, TNF-$\alpha$, and IL-1$\beta$ in the right frontal lobe of rats

To confirm the antioxidant effect of celecoxib, the expression of COX-2 was detected using Western blot. Figure 5 shows that the expression of COX-2 in the tissues around the right frontal lobe injury in rats was significantly increased compared with the sham group ($P < 0.05$). Furthermore, treatment with celecoxib reduced the expression of COX-2 ($P < 0.05$), indicating that celecoxib had an antioxidant effect.

Up-regulation of COX-2 can lead to the release of inflammatory factors such as TNF-$\alpha$ and IL-1$\beta$ (*Kuwano et al., 2004*). In this study, we used ELISA to detect the expression of IL-1 and TNF-$\alpha$ in the right frontal lobe of rats. The results showed that the expression of TNF-$\alpha$ and IL-1$\beta$ was significantly increased after right frontal lobe injury ($P < 0.05$). In contrast, treatment with celecoxib inhibited the change of expression of TNF-$\alpha$ and IL-1$\beta$ ($P < 0.05$). Moreover, celecoxib inhibited the release of related inflammatory factors in the right frontal lobe of rat brains.

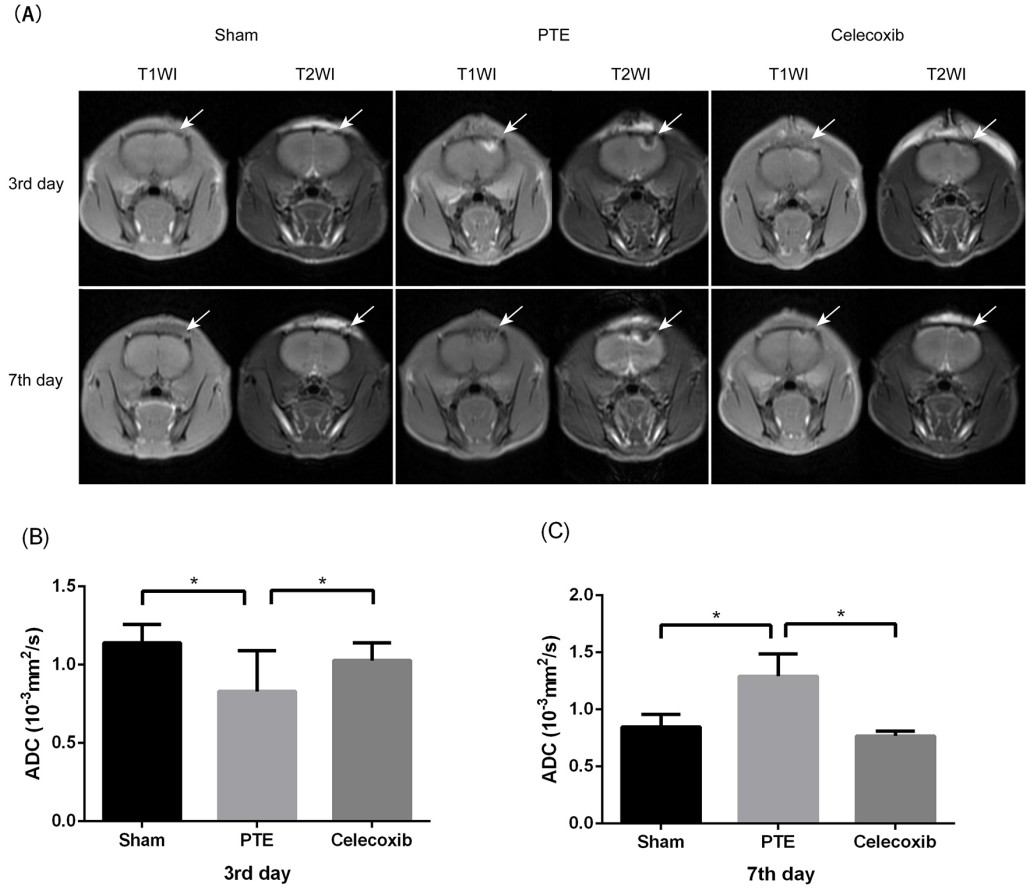

**Figure 3 Change of magnetic resonance imaging (MRI) in post-traumatic epilepsy rats.** (A) shows T1WI and T2WI images of the brain's frontal lobe at different time points after PTE. The arrow indicates damage to the right frontal lobe. (B and C) show different ADC values at the sham, PTE, and celecoxib group lesion on the 3rd or 7th day. Each bar represents the mean ± SD. *$p < 0.05$.

## DISCUSSION

PTE is a common long-term consequence of traumatic brain injury, which is associated with various characteristic pathophysiological changes: focal and diffuse inflammation, neural network remodeling, and subsequent oxidative stress deriving from iron deposition after red blood cell rupture (*Frey, 2003*; *Li et al., 2019*). Notably, antiepileptic drugs are used as a first step in treatment following acute traumatic brain injury. However, research on the prevention of PTE has shown that antiepileptic drugs are limited in reducing the incidence of PTE (*Hakimian et al., 2012*; *Schierhout & Roberts, 2013*). Our study showed the effect of celecoxib on reduced seizure activity, which was observed by behavioral grading using the Racine scale. Pathological outcomes also showed that celecoxib significantly reduced cell death in brain tissue, indicating that celecoxib could effectively reduce the incidence of PTE, especially inhibiting the development of cell death.

In the present study, MRI was used to observe PTE rat models. The MRI examinations were all performed at 3.0T MRI scanners with turbo spin echo (TSE) T1WI, TSE T2WI,

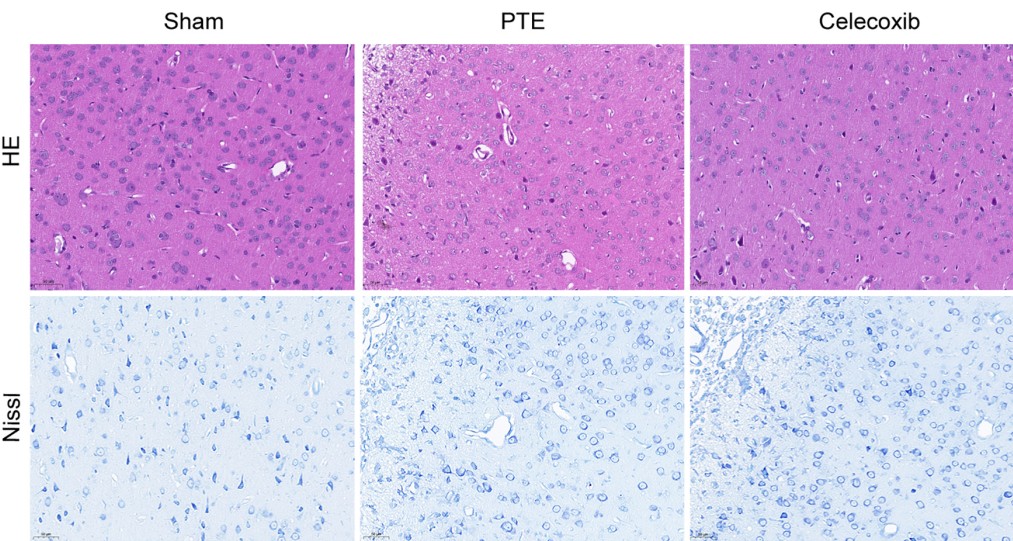

**Figure 4** **Pathological change of post-traumatic epilepsy rat.** On the 7th day following post-traumatic epilepsy of each group of rats, the brains were excised. The PTE group had severe damage (reduced neurons, increased intercellular spaces, condensed cytoplasm, shrunken or dissolved nuclei, and swollen glial cells). In the celecoxib group, a relatively typical structure was observed (HE and Nissl staining, magnification, 200 ×).

and DWI scan sequences. We found cerebral edema in injured brain tissues because there were high signals on T1WI and low signals on T2WI in the injection area of PTE group and treatment group at the same time. Therefore, we concluded that the frontal lobe of the FeCl2-injected rats caused apparent responses to brain damage. In addition, outcomes of MRI on the 7th day showed decreased signal T1WI intensity and low signal T2WI intensity, similar to the corresponding period of clinical brain trauma. Importantly, these results determined that MRI characterization of structural rat brain changes in response to FeCl2-injected post-traumatic epilepsy rat models helps to study the effect of celecoxib treatment of PTE.

From DWI imaging tests, the apparent diffusion coefficient (ADC) reflects the diffusion rate of water molecules within a tissue, which can be used for quantitative analysis. In Capizzano AA's study, ADC was used as an imaging biomarker of PTE to evaluate the condition of epilepsy (*Capizzano et al., 2001*; *Capizzano et al., 2019*), showing that ADC values correlate with cytotoxic edema. On the third day of each successful model establishment, our results showed that quantitatively analysis of ADC values revealed the dispersion degree of molecules of cortex around the injured area of the right frontal lobe, indicating that the ADC value of the PTE group was lower than that of sham and treatment group. Here, injection of iron ions induced inflammation and oxidative reaction in the brain tissues, resulting in the change of ADC value mainly in cytotoxic brain edema. Finally, on the 7th day of each successful model establishment, because the ADC value of the celecoxib group was very close to normal, these data suggested that celecoxib intervention significantly reduced brain edema and improved the condition. All of the above data

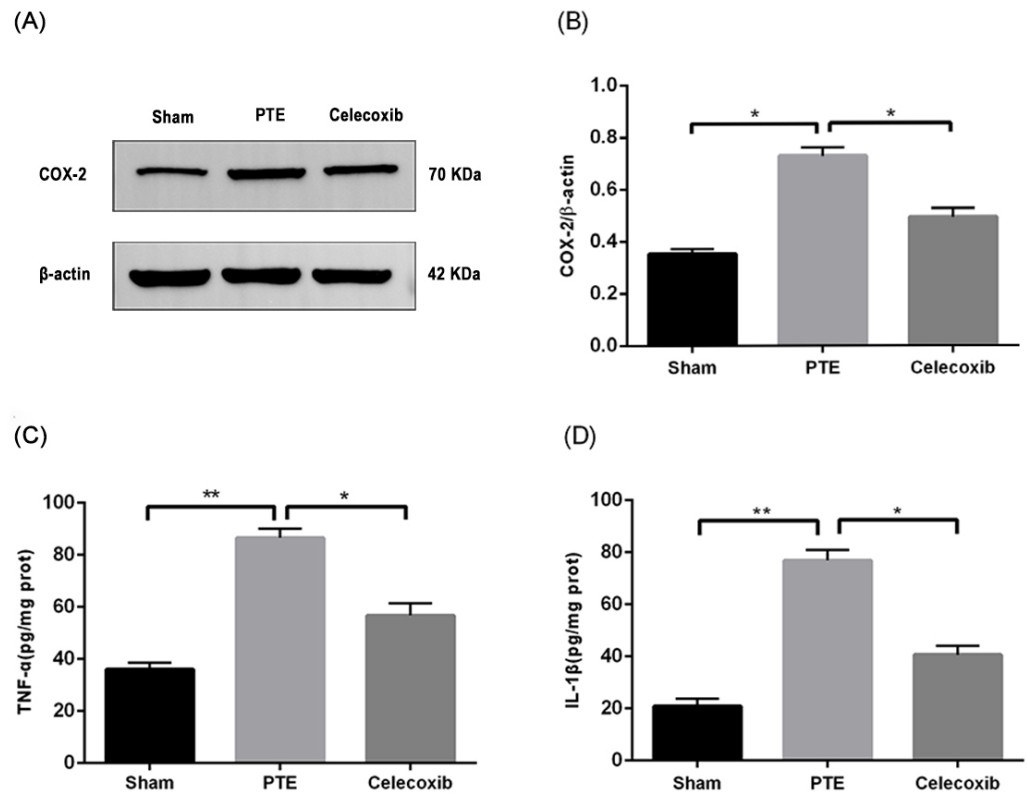

**Figure 5  Expression levels of COX-2, TNF-α, and IL-1 β.** Right frontal lobe tissue proteins from male adult Sprague-Dawley rats are prepared 7 days after PTE. The COX-2 (A) contents were determined by Western blot assay, and the ratios of COX-2/β-actin (B) were calculated based on the densities of the bands. TNF-α (C) and IL-1 β (D) were detected by ELISA. Each bar represents the mean ± SD. *$p < 0.05$; **$p < 0.01$.

indicated that the mechanism of celecoxib's treatment was related to alleviating brain edema.

In response to craniocerebral trauma, the activation of microglia and astrocytes will promote the release of many inflammatory cytokines in the systemic circulation and brain tissue. Inflammatory cytokines lead to neuronal structure damage, synaptic remodeling, excitability enhancement, and then induce epilepsy (*Webster et al., 2017*). IL-1 β has been investigated extensively for its neuroinflammatory role as a studied member of the IL-1 family. Additionally, various brain cells contain the IL-1R, including endothelial, ventricular, astrocyte, and DG neurons (*Schädlich et al., 2022*). In the central nervous system, IL-1β exerts inflammatory effects through the IL-1/IL-1R signaling pathway (*Liu et al., 2019*). Researchers have also provided a vast amount of evidence supporting the involvement of IL-1β in epilepsy (*Mukhtar, 2020*). Tumor necrosis factor (TNF)-α is also known to stimulate the proliferation of astrocytes; therefore, it may play a role in post-traumatic epilepsy (*Wang et al., 2018a*; *Wang et al., 2018b*). Notably, TNF-α is involved in epilepsy by releasing glutamate, mediating the downregulation of connexin43, and controlling synaptic transmission (*Ravizza & Vezzani, 2018*). Several studies have

also demonstrated that high-dose TNF-$\alpha$ can significantly induce seizures, and TNF-$\alpha$ antagonists can prevent seizures (*Arulsamy & Shaikh, 2020*). Moreover, *De Simoni et al. (2000)* found that the induction of spontaneously recurring seizures in rats involves the activation of inflammatory cytokines such as IL-1$\beta$ and TNF-$\alpha$ in the hippocampus. These changes may play an active role in the hyperexcitability of the epileptic tissue. These studies suggest that detecting IL-1$\beta$ and TNF-$\alpha$ can be diagnostic indicators for epilepsy and brain injury. In this study, we found that levels of TNF-$\alpha$ and IL-1$\beta$ were significantly higher than those in the control group, indicating that TNF-$\alpha$ and IL-1$\beta$ played a vital role in the pathogenesis of epilepsy.

When moderate or severe craniocerebral trauma occurs, seizures can be triggered by an inflammatory reaction (*Vezzani, Friedman & Dingledine, 2013*). COX-2 is rapidly induced by PTE, resulting in a subsequent release of prostaglandins (PGs), potent mediators of inflammatory cytokines such as TNF-$\alpha$ and IL-1$\beta$ (*Rawat et al., 2019*). Additionally, inflammation often leads to an increase in blood–brain barrier (BBB) permeability. Neurotransmitters and neurons are affected when the BBB becomes damaged, resulting in inflammatory factors entering the brain. As a result, the damaged neurons activate more microglia cells, causing the disease to worsen in a vicious cycle (*Galea, 2021*). A COX-2 inhibitor can reduce oxygen free radicals, nitric oxide, and interleukin-1 by inhibiting TNF-$\alpha$ and PG synthesis, which improves neuroinflammatory reactions (*Desai, Prickril & Rasooly, 2018*). In this study, we revealed that cyclooxygenase-2 inhibitors inhibit COX-2, TNF-$\alpha$, and IL-1$\beta$ levels, producing a significant anti-inflammatory effect at the early stage of PTE. Furthermore, our study showed that COX-2 plays a vital role in neuroinflammatory responses and regulates TNF-$\alpha$ and IL-1$\beta$. However, its specific mechanism needs further investigation.

## CONCLUSIONS

In conclusion, the therapy with celecoxib, a selective COX-2 inhibitor, is suggested to have a significant therapeutic effect in the PTE model through its anti-inflammatory effect. We used MRI to determine the ADC value of the area around the injection site of the right frontal lobe, which helps assess the disease's development.

### Abbreviations

| | |
|---|---|
| **ADC** | apparent diffusion coefficient |
| **HE** | hematoxylin-eosin |
| **COX-2** | cyclooxygenase-2 |
| **IL-1** | interleukin-1 |
| **TNF-$\alpha$** | tumor necrosis factor$\alpha$ |
| **IL-1$\beta$** | interleukin-1$\beta$ |
| **NSAIDs** | Non-steroidal anti-inflammatory drugs |

### Funding

The authors received no funding for this work.

### Competing Interests

The authors declare there are no competing interests.

### Author Contributions

- Lei Chen conceived and designed the experiments, prepared figures and/or tables, authored or reviewed drafts of the article, and approved the final draft.
- Qingsheng Niu conceived and designed the experiments, analyzed the data, prepared figures and/or tables, and approved the final draft.
- Caibin Gao conceived and designed the experiments, performed the experiments, analyzed the data, authored or reviewed drafts of the article, and approved the final draft.
- Fang Du performed the experiments, analyzed the data, prepared figures and/or tables, authored or reviewed drafts of the article, and approved the final draft.

### Animal Ethics

The following information was supplied relating to ethical approvals (i.e., approving body and any reference numbers):

Shizuishan First People's Hospital's Institutional Animal Care and Use Committee

### Data Availability

The raw data is available in the Supplemental Files.

### Supplemental Information

Supplemental information for this article can be found online at http://dx.doi.org/10.7717/peerj.16555#supplemental-information.

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
