# Peer review of "Celecoxib treatment alleviates cerebral injury in a rat model of post-traumatic epilepsy"

_PeerJ, doi:10.7717/peerj.16555_

## Round 0.1 · original submission · Major Revisions

This study showed that celecoxib has a significant therapeutic effect on rat PTE model through its anti-inflammatory effect, of which the topic is interesting.
Overall, I think the experimental design is OK, but more decriptions should be provided, such as the aim for each work.

There are so many grammar errors and typos. When presenting the contents of images and tables in English, the present simple tense should be used instead of the past tense.

Please make full revisions according to reviewers' comments.

**Language Note:** The Academic Editor has identified that the English language must be improved. PeerJ can provide language editing services - please contact us at copyediting@peerj.com for pricing (be sure to provide your manuscript number and title). Alternatively, you should make your own arrangements to improve the language quality and provide details in your response letter. – PeerJ Staff

·

Basic reporting

Comments

Title: Celecoxib treatment alleviates cerebral injury in a rat model of post-traumatic epilepsy

The paper is accepted with major revision and correction

Abstract:
Please, Write the group classification in methods.
Clarify this abbreviation “3.0TMRI”.
In this line “expression of COX-2, interleukin-1³ (IL-1³), and tumor necrosis factor-alpha (TNF-³) in brain”, delete the full term and use only the abbreviation you mentioned before “TNF-³”.
Write a brief recommendation after the conclusion.

Introduction
In this line “astrocytes, and glucose metabolism dysregulation (Sharma et al., 2019, Webster et al., 2017).” and other similar parts of paper, write the references from old to recent to be “astrocytes, and glucose metabolism dysregulation (Webster et al., 2017; Sharma et al., 2019).”.
In this line “evidence in vivo experimental models suggested there is a rapid release of inflammatory”, add word “that” to be “evidence in vivo experimental models suggested that there is a rapid release of inflammatory”.
Please, clarify this abbreviation “NSAIDs”.
No need for this “Based on rare treatment of PTE, in this study, a rat model that replicates clinical PTE was achieved by injecting ferrous chloride into the frontal lobe. We treated rats with celecoxib, observed the level of epilepsy and investigated pathological changes in the injury site of the brain, mainly including neuronal cells and glial cells. In addition, the brain structure was examined after 3 and 7 days using T2wt MRI, T1-Gd MRI, and DTI and compared these MRI features such as tissue ADC, which aims to investigate application of Nuclear Magnetic Resonance (NMR) and therapeutic effects of celecoxib on brain injury in PTE rats.”, Please, write the aim of work in brief.

Materials and Methods
You mentioned “we targeted FeCl2 into the right frontal lobe”, please, add a photography to show it.
In this line “On the 3rd and 7th days after the model was delivered, Magnetic resonance imaging (MRI)”, Please, not repeat the full term once you mentioned the abbreviation before.
For each paragraph of methods, please, add a reference.
For Histopathologic Processing, you can add this reference:
Hegazy R, Hegazy A. Hegazy’ simplified method of tissue processing (consuming less time and chemicals). Ann Int Med Den Res. 2015;1(2):57.

Results
In the first paragraph, change the verbs to the past, not the present or future.

Conclusions
The first line may be changed to be “In conclusion, the therapy with celecoxib, a selective COX-2 inhibitor, is suggested to have a”.

Please, delete these repeated references:
17. Webster KM, Sun M, Crack P, O’Brien TJ, Shultz SR, Semple BD. 2017. Inflammation in epileptogenesis after traumatic brain injury. J Neuroinflammation 14(1):10 DOI 10.1186/s12974-016-0786-1.
26. Rawat C, Kukal S, Dahiya UR, Kukreti R. 2019. Cyclooxygenase-2 (COX-2) inhibitors future therapeutic strategies for epilepsy management. J Neuroinflammation 16(1):197. DOI 10.1186/s12974-019-1592-3.

Figures
Please, write more details in the legend of all figures & write the verbs in the present, not in the past.
Figure 3: Write the type of stain and magnification at the end of legend.

Experimental design

decent.

Validity of the findings

decent.

Additional comments

None.

Reviewer 2 ·

Basic reporting

In the manuscript titled “Celecoxib treatment alleviates cerebral injury in a rat
model of post-traumatic epilepsy”, the author proved that celecoxib has a significant therapeutic effect on rat PTE model through its anti-inflammatory effect. This is a good study, but some questions still need to be answered:

Experimental design

1.In Figure 1, what time points did the Racine score of PTE group significantly different from that of celecoxib group?
2. Before comparing the Racine scores of PTE group and celecoxib group, would it be better to explain the Racine score?
3.It is better to use full names for academic terms that first appear in the manuscript, and to use abbreviations when they appear again later.

Validity of the findings

4. The manuscript would benefit from additional language editing and grammar corrections.

·

Basic reporting

no comment

Experimental design

no comment

Validity of the findings

no comment

Additional comments

The authors present a study of the effects of celecoxib treatment on post-traumatic epilepsy. In this work, the authors revealed that COX-2 selective inhibitor celecoxib significantly reduced the cerebral injury through anti-inflammatory effect, and effectively alleviated the degree of seizures. However, before the acceptation for publication, several problems should be eliminated.
1. This paper only observed the early seizures within 7 days after successful modeling, and did not observe the subsequent late seizures, which is an obvious defect. Clinically, seizures after traumatic brain injury can be divided into early seizures and late seizures. For early seizures, it is generally believed that they are caused by pathophysiological factors such as brain edema, inflammatory response. Therefore, this seizure is usually called a seizure, rather than an epileptic seizure
2. This paper mainly studied the effects of celecoxib on cerebral injury such as brain edema, neuroinflammation and cell death, instead of the seizure frequency, seizure duration, seizure circle of epilepsy. Authors need to keep the consistent of “alleviates cerebral injury” in the title and “effect of celecoxib on early epilepsy” in Objective of Abstract part.
3. The information on 20% of patients after traumatic brain injury developing epilepsy is misleading. It is 20% of patients in the group with SEVERE TBI only. Please refer to the works of Pitkanen et al
4. There is no information on how the tissue samples were obtained.
5. Last sentence, paragraph 2 of Introduction: I believe there are appropriate models yet no-one studied the effects of celecoxib in those.
6. Discussion should be shortened by ½ page and focused on current findings.
7. Statistic difference should be shown in Figure 1.
8. Language modification is needed, especially the abstract.

---

## Round 0.2 · Minor Revisions

This manuscript has been improved. As suggested by reviewers, there are several minor concerns.
1. delete this abbreviation “(SD)” used for “Sprague-Dawley (SD)” ;
2. Figure legends should be reconstructed. In figure 3 legend, authors do not need to mention "figure 3A". "(A)" is better.

·

Basic reporting

Only the authors may delete this abbreviation “(SD)” used for “Sprague-Dawley (SD)” throughout the paper. This is because it is not used. It may also mislead other abbreviations used for "standard deviation (SD)".

Experimental design

The paper is accepted.

Validity of the findings

The paper is accepted.

Additional comments

No further comments.

·

Basic reporting

no

Experimental design

no

Validity of the findings

no

Additional comments

Figure legends should be reconstructed normatively.

---

## Round 0.3 · accepted · Accept

Authors have made full revisions and addressed all the comments. This paper can be accepted for publication.